# Diagnostics of Internal Defects in Composite Overhead Insulators Using an Optic E-Field Sensor

**DOI:** 10.3390/s24051359

**Published:** 2024-02-20

**Authors:** Damiano Fasani, Luca Barbieri, Andrea Villa, Daniele Palladini, Roberto Malgesini, Giovanni D’Avanzo, Giacomo Buccella, Paolo Gadia

**Affiliations:** RSE SpA—Ricerca sul Sistema Energetico, Via Raffaele Rubattino, 54, 20134 Milano, Italy; luca.barbieri@rse-web.it (L.B.); andrea.villa@rse-web.it (A.V.); daniele.palladini@rse-web.it (D.P.); roberto.malgesini@rse-web.it (R.M.); giovanni.davanzo@rse-web.it (G.D.); giacomo.buccella@rse-web.it (G.B.); paolo.gadia@rse-web.it (P.G.)

**Keywords:** composite insulator, live-line diagnostics, electro-optic sensor

## Abstract

Composite insulators for high-voltage overhead lines have better performances and are lighter than traditional designs, especially in heavily polluted areas. However, since it is a relatively recent technology, reliable methods to perform live-line diagnostics are still under development, especially with regard to internal defects, which provide few external symptoms. Thermal cameras can be employed, but their use is not always straightforward as the sun radiation can hide the thermal footprint of internal degenerative effects. In this work, an optical E-field sensor has been used to diagnose the internal defects of a set of composite insulators (bandwidth 200 mHz–50 MHz, min. detectable E-field 100 V/m). Moreover, a modelling activity using finite elements has been carried out to identify the possible nature of the defects by comparing experimental E-field profiles with those simulated assuming a specific defect geometry. The results show that the sensor can detect the presence of an internal defect, since its presence distorts the E-field profile when compared to the profile of a sound insulator. Moreover, the measured E-field profiles are compatible with the corresponding simulated ones when a conductive defect is considered. However, it was observed that a defect whose conductivity is not at least two orders of magnitude greater than the conductivity of the surroundings remains undetected.

## 1. Introduction

This paper deals with the use of a fully dielectric, electro-optic E-field sensor to diagnose internal defects in composite line insulators. Composite line insulators are, to date, a viable alternative to conventional insulators for transmission and distribution networks [1]. The interest in this type of insulators is due to their generally superior performance compared to glass or ceramic cap and pin insulators, especially in highly polluted environments as a result of their hydrophobicity [2]. In addition, composite insulators are significantly lighter and, therefore, easier to transport and install. The typical structure of a composite insulator consists of three elements [3]:a fiber-reinforced plastic (FRP) rod essentially resistant to tensile mechanical stresses;a housing made of polymeric material (e.g., silicone rubber) with lamellar geometry;two metal terminals at the ends.

Composite insulators are also subject to ageing and degradation: like traditional insulators, they are exposed to external contamination, which may create conductive patterns on the silicone surface with consequent corona discharges, the degradation of the material (which loses its hydrophobic properties), and flashovers [4]. Several standards have been established throughout the years to test the performances of these insulators when subjected to ageing (see, for example, [5]) However, the most critical aspect concerning the ageing of composite insulators is the live-line diagnosis of internal defects. Pre-existing defects, introduced, for example, during the manufacturing process between the rod and the sheath can promote partial discharge (PD) phenomena, leading to interface breakdown [6,7] with the consequent penetration of moisture and external pollutants. This can result in flashunders and the degradation of the rod, which eventually leads to its mechanical failure (e.g., brittle fracture of the FRP rod [8]). However, there are some cases, as described in this work, where the observed defects do not appear to be related to the PD activity.

Internal degradation phenomena can be initiated and continue over the years without the insulator appearing damaged upon the initial external visual inspection. Several methods have been developed to detect internal and external defects for composite insulators in operation. UV detection by means of special cameras is particularly suited for localizing the initial corona activity, but it is not able to detect PDs occurring under the insulator surface [1,9]. Antennae have been used to detect electromagnetic waves radiated as a consequence of PDs [10,11], even though correlating the resulting wave spectrum to the kind of defect (and to whether it is internal or external) is not a simple task. It is clear, however, that such techniques cannot be used to identify defects that do not induce PDs. Infrared (IR) thermography can be used to detect conductive defects, as they generally correspond to areas with a higher temperature. It is worth emphasizing that, in this paper, the term “conductive” is meant in relation to the insulator material; it refers to the material having higher conductivity values than those of a non-aged insulating material, which are still much lower than those of a metal. However, this technique is not very effective for those defects that cause small variations in temperature. Especially, thermography is not easily applicable in direct sunlight [7]. Another class of live-line diagnostics methods involves the measurement of the electric field profile near the insulator. Measurements are obtained by scanning the E-field along a line parallel to the axis of the insulator at a distance of a few tens of centimeters using a suitable probe. Internal defects tend to distort the electric field; therefore, their presence can be assessed by comparing the E-field profile of the insulator being tested to the profile of a healthy one. As reported in a recent review [12], different technologies have been developed to perform E-field measurements of line insulators. Force-based sensors exploit the Coulomb force on a charge located on a flexible element when an external E-field is applied and on the detection of the consequent displacement of such element. Since the forces are very small, the readout is challenging, and no commercial devices are currently available. Sensors based on electrostatic induction are commercially available, and, depending on the configuration, they can be suitable for both DC and AC E-fields. The measurement principle is based on the detection of the current that is induced at its sensing electrodes by the variation of the E-field flux seen by the electrodes themselves. Electro-optic (EO) techniques are based on the property of certain dielectric materials to change their optical properties (e.g., absorption and refractive index) when an E-field is applied. Since sensors based on EO elements are usually fully dielectric, they have limited influence on the electric field to be measured. Within this class of methods, four technologies have mainly emerged: optical fibers as the sensing element, interferometry, resonating cavities, and polarimetry measurements. Optical fiber-based methods, that rely on the change in optical absorption due to an external E-field, can be quite ineffective at high frequencies given their usually large dimensions. However, recent developments have achieved extremely compact designs, which are also promising in terms of bandwidth [13]. Interferometric methods [14,15,16] are based on the splitting of a laser beam into two paths, travelling along two different arms. The properties of the light guides are chosen so that, as the light is recombined, interference effects occur, leading to the possibility of detecting the electric field. Sensors based on this technique have a large bandwidth and high sensitivity [17]. On the other hand, detecting the direction of the electric field remains very difficult. Measuring techniques based on resonant cavities use transducers obtained by creating an interface between an optical fiber and an EO crystal. As the refractive index of the crystal changes with the external electric field, the amount of light exchanged between the crystal and the fiber changes along with the total amount of light transmitted through the fiber [12,18].

In this work, we perform polarimetry measurements by means of an electro-optic sensor, which was already developed and characterized in [19]. The electro-optic crystal changes its refractive index according to the electric field: as a light beam passes through this element, its polarization state is changed depending on the field strength. If polarimetric measurements of the exiting beam are carried out, this technique makes it possible to obtain the amplitude of the electric field along one direction with a relatively large rejection of the orthogonal components. As already stated, and contrary to other technologies [20], this kind of sensor has the advantage of being fully dielectric, thus introducing negligible perturbations of the electric field and reducing the threats posed to live-line workers performing the measurements induced by the presence of metallic parts [21]. Moreover, these sensors have a bandwidth of tens of MHz, which makes them suitable for measuring the E-field in normal operation conditions, as well as for PD detection.

In this work, the sensor has been used to diagnose internal defects of composite insulators during normal operation: the purpose is to evaluate the response of the sensor to the presence of defects and, thus, to assess its ability to discern, based on the trend of the vertical electric field profile, healthy insulators from defective ones. Moreover, a modelling activity was carried out using a finite element simulation software (COMSOL Multiphysics^®^ 6.1), and a sensitivity analysis was performed to observe the effect of the simulated defect characteristics (actual location, size, and electrical properties are unknown) on the E-field profile. The goal of such an activity is to identify the possible properties of the defect, i.e., the set of physical and geometrical properties that give rise to a simulated E-profile as close as possible to the experimental one. On the other hand, the modelling activity is aimed at identifying which defect properties (and their range of variation) have the most significant influence on the distortion of the electric field and, thus, what kind of defects are “visible” to the sensor. The paper is structured as follows:A description of the sensor and the other relevant piece of experimental equipment is provided, together with the test procedure and the assumptions behind the finite element model;The results of the experimental activity are presented and discussed;The results of the modelling activity are discussed and compared with the results of the tests.

## 2. Materials and Methods

All tests were carried out in a 2-day campaign inside a Faraday cage, in order to shield the experimental equipment from environmental disturbances. A set of four identical composite insulators for high-voltage applications has been used for this activity. These are made from a 1 m-long fiberglass rod with a diameter of 21 mm enveloped in a silicone rubber housing characterized by 19, 122 mm diameter sheds (major sheds) interposed by 18, 90 mm diameter sheds (minor sheds), as reported in Figure 1. For clarity of the upcoming discussion, major sheds have been numbered from the 1st to the 19th starting from the high-voltage terminal.

Among the insulators provided, two had supposedly no defects, as they were of recent manufacture and were never operated on a real line. The remaining two—which experienced ageing as they were operated on actual lines—had internal defects, according to the manufacturer. The approximate location of these defects was identified during testing by means of a Ti400 IR camera by Fluke Corp. (Everett, WA, USA) All insulators were labelled using appropriate numbering; specifically, defect-free insulators were labelled #2 and #6, while defective insulators were labelled #1 and #5. Figure 2 compares thermographic images of a healthy insulator and a defective one; the latter can be recognized by the presence of a region with temperature higher than the rest of the insulator. To assess whether the insulators were affected by PD activity, preliminary measurements with the MPD 600 by Omicron electronics GmbH [22,23] (Klaus, Austria) were carried out. No PD activity was detected.

The insulator was suspended from the ceiling of the Faraday cage—which constitutes the ground terminal—and was connected, at the lower end-fitting, to a transformer (HV terminal), which provided a voltage of 70 kV AC at 50 Hz (rms).

To measure the electric field, the fully dielectric optical sensor described in [19] was used. A linearly polarized laser beam is injected in a polarization-maintaining (PM) optical fiber which connects the laser source with the sensor. Several different laser sources could be used: both He-Ne sources and laser diodes with a reference power ranging from 1 mW to 5 mW were available. In this set of tests, a 5 mW He-Ne source at 633 nm by Thorlabs (Newton, NJ, USA) was used. A quarter-wavelength plate is placed at the beginning of the sensor assembly to change polarization from linear to circular. The laser beam, then, passes through a non-linear electro-optic crystal which, when an electric field is applied, modifies the light polarization state. Finally, the beam is split by a couple of Wollaston prisms to obtain the polarization state in four directions at 0, 45, 90, and 135 degrees with respect to the crystallographic axes of the electro-optic crystal. The four laser beams, emerging from the sensor, are collected by four multi-mode fibers and sent to a signal converter and digitalization unit. This particular arrangement makes it possible to keep the laser source and the electronics far away from the measurement region, which is close to objects energized at high voltages.

By measuring the relative intensities of the laser beams of the four fibres, the intensity and the direction of the electric field vector can be determined as described in [19]. More precisely, it is possible to determine the electric field vector on a plane perpendicular to the laser beam in the sensor.

In this work, a slightly simplified version of the sensor was used, as depicted in Figure 3b: a single Wollaston prism was used and only the polarization state in the directions at 0 and 90 degrees were measured. This made it possible to measure the electric field component aligned with the direction at 0 degrees from the crystallographic axes of the crystal. This arrangement is cheaper, smaller, and easier to assemble than the version described in [19], thus being more suitable for this particular application. Indeed, we show that the measurement of a single electric field component is sufficient for performing a proper diagnostic. Like any other optical E-field sensor, this one has a very large bandwidth, ranging from 200 mHz to 50 MHz. The band is influenced by the design of the electronics and the required signal-to-noise ratio.

The sensor was placed and moved near the insulator using a proper handling system consisting of a base connected to two 4 m-tall cylindrical poles. A crossbar is mounted on the poles’ upper ends to enhance the rigidity of the system and to keep the correct distance between the poles. The optical element is fixed on a sled made of a plexiglass plate fitted with four sleeves at its corners, which allow it to slide vertically along the poles, as represented in Figure 4. The sled is pulled through a pulley mounted on the crossbar by means of a winch installed at the base of the frame operated via remote control. The electro-optical sensor is fixed to the sled in such a way as to detect the component of the electric field that forms an angle of 45° with respect to the vertical direction.

All elements of the setup (apart from the winch control system) are made of dielectric materials to avoid introducing perturbations in the electric field. The following procedure was adopted to measure the vertical profile along the insulator:Voltage was applied to the insulator;The optical element was positioned at a specific height (midway between two consecutive major sheds) and the signal was acquired for a time of 100 ms;The sensor was moved upwards to the location corresponding to the adjacent couple of sheds and the acquisition procedure was repeated.

The scan of the E-field profile began at a height corresponding to the 1st–2nd shed (about 2.8 m above the floor) and ended at a height corresponding to the 17th–18th shed (3.6 m above the floor), for a total of 9 acquisition points along the vertical axis and a corresponding spatial resolution of 0.1 m. In all tests performed, the handling system was positioned to ensure a distance of 0.13 m between the sensor and the axis of the insulator. Since tests were carried out during a two-day campaign, the ones performed on an insulator the first day were repeated under the same conditions the second day, in order to take into account possible errors due to a different positioning of the handling system when comparing the results for different insulators. Finally, to verify the possible effects of the thermal transient, tests were performed at different elapsed times since energization. Specifically, two of the four insulators—one healthy and one defective—were tested both immediately after applying voltage (“cold” test) and 30 min after applying voltage (“hot” test), a reasonable elapse time to ensure steady-state operation from a thermal perspective. Table 1 summarizes the conditions of each test.

As already mentioned in the introduction, an axisymmetric 2D geometry model of the insulator was constructed using a finite element simulation software (COMSOL Multiphysics^®^). This reproduces the test conditions to which the real insulator is subjected and returns the electric field distribution over the whole domain. The E-field component in the 45° direction was computed according to Equation (1) along a vertical line at a distance of 0.13 m from the insulator axis (the same as the sensor in the experimental setup):(1)$$E_{45}(z)=\left[E_{r}(z)\cos\left(\pi /4\right)+E_{z}(z)\sin\left(\pi /4\right)\right]/4.81,$$
where $E_{r}$ and $E_{z}$ are the components in the radial and axial directions of the electric field in air, respectively, while 4.81 is a conversion factor between the electric field in air and the electric field inside the electro-optical crystal. The factor was obtained by matching the simulated profile in the absence of defects and the experimental profile of a healthy insulator. All simulations were carried out using the Electric Currents (ec) interface of the AC/DC Module, which, given the small current densities involved, is an excellent approximation of the physics of the problem. The conductivity and permittivity of the insulator materials used in the simulation are reported in Table 2, and are in line with what can be found in the literature [24] and among commercial products [25]. However, it is worth noting that a high variability was found, especially in the conductivity of these materials. The dependence of the solution on the variability of properties will be investigated in future works.

The actual nature, position, and dimensions of the defective region of insulators are unknown. For the simulations performed in this work, it was assumed that the defect consists of a delamination at the interface between the fiberglass rod and the silicone housing, which is a rather frequent occurrence among the possible degradation phenomena. Indeed, temperature cycles (e.g., night/day) can introduce mechanical stress at the interface between materials with different thermal expansion coefficients: over time, this may result in adhesion failure at the interface and in the formation of a gap between the two components, which is characterized by dielectric and resistive properties different from the surrounding material. It was initially assumed that the measured increase in electric field strength could be due to the presence of liquid water in the delamination region, due to air infiltrations and subsequent deposition of condensation (given the right temperature and humidity conditions). Therefore, assuming that the defect has the electrical properties of water (*ε_r,d_* = 81, *σ_r,d_* = 5.5 × 10^−6^ S/m) and a thickness of 0.1 mm, the effect of its length and location on the electric field profile was studied (see Table 3).

Finally, a sensitivity analysis was carried out for a fixed geometry to check the influence of the electrical properties of the defect on the vertical electric field profile. The range of variation of parameters is given in Table 4.

## 3. Results and Discussion

### 3.1. Experimental Results

Figure 5 shows the electric field profile, obtained from tests on the four insulators, versus the axial co-ordinate (i.e., the height of the sensor above the ground during a measurement campaign).

The difference between the profile of healthy insulators (#2 and #6) and that of defective ones (#1 and #5) is quite evident. In the former case, the intensity of the E-field decreases monotonically as the axial co-ordinate increases. In the latter case, the electric field value shows a maximum whose position depends on the insulator: #1 has a maximum between the 3rd and 4th shed, while #5 has a maximum between the 5th and 6th shed.

Insulators #1 and #6 were subjected to both “cold” and “hot” tests, represented in Figure 5 with dashed and dotted lines, respectively. The resulting profiles measured in “hot” conditions closely follow their “cold” counterparts. This is especially evident for insulator #1 (tests #1-d1-C and #1-d1-H), whose hot and cold profiles are overlapping. On the other hand, the small differences observed for insulator #6 (#6-d2-C and #6-d2-H) might be attributed to errors in the vertical positioning of the optical element, rather than to the effect of the temperature. Similarly, the two tests performed one week apart on insulator #1 under the same conditions (#1-d1-C and #1-d2-C) resulted in two profiles sharing similar characteristics (e.g., the position of the maximum E-field strength) despite small differences. Given the short period between tests, it is likely that the observed discrepancy is due to a slightly different positioning of the sensor-handling system or of the insulator itself in the tangential direction (there is no guarantee that the defect is axisymmetric), as well as to the vertical positioning of the probe. Overall, the present experimental setup was capable of ensuring good repeatability, since multiple tests on the same specimen resulted in profiles with similar features, despite the small differences likely due to positioning, whose effect is discussed in Section 3.2.

### 3.2. FEM Model

Here, the results of finite element modelling of the electric field distribution surrounding an insulator are reported, for both a new and a defective insulator.

As an example, Figure 6 represents the electric field distribution (modulus) inside and outside a new insulator (a) and an insulator with a defect spanning a height of 210 mm (b), between the metal fitting and the fourth to fifth shed. It is evident that the presence of the defect results in a reduction of the electric field strength inside the insulator rod along the axial span of the defect and, at the same time, in an increase outside the insulator, which is particularly evident in the region adjacent to the defect. Figure 7 shows, with solid lines, the simulated vertical profile of the E-field of a new insulator and that of a defective insulator for different values of the axial dimension *H_d_* of the defect and of its position *z_d_* (i.e., the positions of the lower end of the defect with respect to the floor, where z = 2715 mm is the position of the metal fitting). The dashed lines represent the experimental results. In the case of a healthy insulator, the plots show that the simulated profile, in the absence of defects, closely follows the monotonically decreasing trend of the corresponding measured profile, confirming the validity of the numerical–experimental approach used. Regarding the influence of the defect geometry, Figure 7a shows that a conductive defect positioned exactly at the HV terminal (*z_d_* = 2715 mm) results in a modelled E-field profile with a trend similar to the measured ones. Moreover, as its axial dimension increases, the maximum of the electric field strength shifts to larger axial co-ordinates, while undergoing a slight decrease. Considering the position of the maximum, the *H_d_* = 210 mm profile could be representative of insulator #1, while the *H_d_* = 265 mm profile could represent #5.

A similar behavior can also be observed in Figure 7b, where the simulated defect is positioned 15 mm from the HV terminal: the same shift of the maximum as the dimension increases is observed. However, the curves seem less representative of the experimental results (significant decrease in the value of the maximum). Figure 7c depicts a situation that is not representative of the problem but lends itself to interesting considerations. Indeed, it shows that a defect relatively far from the high-voltage terminal (135 mm, in this case) would cause a small distortion of the electric field profile, as already evidenced, for instance, in [9]. This implies that such defects could remain undetected, unless the number of measurement locations along the insulator is increased to achieve a better spatial resolution.

Finally, Figure 8 shows a simulation of the effect of a probe positioning error in the radial direction. For this purpose, the simulated case, represented in Figure 6, was taken as a reference (*z_d_* = 2715, *H_d_* = 210 mm). The vertical profile of the electric field was computed in three different radial positions *d* with respect to the insulator, i.e., at the nominal sensor distance in the experimental setup (*d* = 130 mm, solid blue line) and at *d* = 125 and *d* = 135 mm (red and purple solid lines, respectively). The last two were chosen to simulate a sensor positioning error of ±5 mm in the radial direction, which corresponds to less than 4% of the nominal distance. For the sake of comparison, the E-field profiles obtained during tests #1-d1-C and #1-d2-C were also represented: both of them refer to insulator #1 and were obtained one week apart. Between the two measurements, the experimental equipment was moved, and then set up again, with all possible consequences on its positioning accuracy. The results suggest that the variation in the measured intensity of the electric field is compatible with a different horizontal position of the sensor. However, it is also worth noting that the differences between the two experimental tests could be attributed to several other causes (vertical positioning of the sensor, different angular position of the insulator, etc.), as already pointed out in Section 3.1.

#### Influence of the Defect Electric Properties

To conclude, taking as reference a defect geometry that can reasonably be considered representative of one of the tested insulators (*z_d_* = 2715, *H_d_* = 210 mm), here, the results of the sensitivity analysis on the influence of the electrical properties of the defect on the E-field profile are reported. The plots in Figure 9 represent the electric field profile for different values of the defect conductivity and for two extreme values of its permittivity. It is evident that, given the same *ε_r,d_*, the electric field profile progressively deviates from the healthy insulator condition as the conductivity deviates from that of the standard insulator materials. On the other hand, no significant dependence on the permittivity of the defect was found: this is probably due to the extremely small size assumed for the defect (thickness: 0.1 mm), such that even a large decrease in the electric field inside the defect does not significantly affect the measured/simulated electric field outside of it.

The sensitivity of the electric field profile as a function of the conductivity of the defect was quantified by computing, for each profile k, the average difference of E-field intensity along the axial co-ordinate between said profile and the profile k+1 obtained for a conductivity $\sigma_{d,k+1}=10\cdot \sigma_{d,k}$. This was normalized to the average electric field strength value of the profile k, namely:(2)$$norm_{k}=\frac{\int_{z_{i}}^{z_{f}}\left|E\left(z;\sigma_{k+1}\right)-E\left(z;\sigma_{k}\right)\right|dz}{\int_{z_{i}}^{z_{f}}\left|E\left(z;\sigma_{k}\right)\right|dz},$$
where $z_{i}$ and $z_{f}$ are the boundaries of the scanning length. This quantity is represented in Figure 10 as a function of electrical conductivity. The figure shows that the conductivity variation might have a measurable effect on a limited range (indicatively between 10^−12^ S/m and 10^−9^ S/m). This means that defects whose conductivity is not at least two orders of magnitude greater than that of the silicone housing (three orders greater than that of the rod) are unlikely to be detected with this experimental setup. On the other hand, defects whose conductivity exceeds 10^−9^ S/m can be easily detected but are indistinguishable from each other.

## 4. Conclusions

In this work, an E-field electro-optic sensor has been presented and tested to perform line insulator diagnostics. The results of an experimental campaign on a set of insulators have shown that the sensor can easily highlight the E-field distortion caused by a defect close to the HV terminal. However, the system is sensitive to the sensor position; therefore, special care in positioning the EO element will be required, especially in future field campaigns. For this purpose, solutions similar to the ones already implemented in commercial products can be adopted: for example, the sensor can be mounted on a sled specially designed to fit on the insulator side and that can be handled by a lineman, as seen in [26]. Simulations using finite elements have shown that the modelled profile closely follows the corresponding measured profile in the case of a sound insulator, i.e., when all the physical parameters are known. In the case of a defective insulator, the simulations for an interfacial delamination, having the electric properties of water and positioned close to the HV terminal, have resulted in E-field profiles similar to the corresponding experimental ones. The simulation has also evidenced that defects far from the HV terminal may be difficult to detect, given the modest distortion of the E-field they can cause. Moreover, defects whose conductivity is not at least two to three orders of magnitude greater than the conductivity of the insulator materials are also invisible to the sensor, regardless of the permittivity value.

In order to improve the quality of simulations, tests to characterize the electrical properties of the insulator material, namely, volume and surface resistivity and permittivity, will be performed. In particular, surface resistivity measurements on samples exposed to pollution in a controlled environment would allow us to take into account the effect of the external pollution on the distortion of the measured electric field. Moreover, the defective insulators will be taken apart and analyzed to assess whether the assumptions on defect geometry and physical properties were correct.

## Figures and Tables

**Figure 1 sensors-24-01359-f001:**
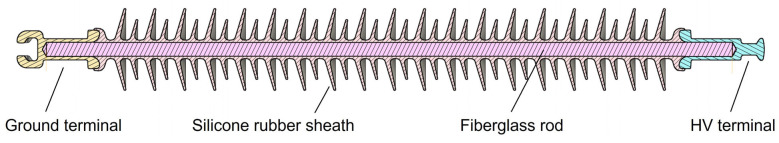
Section view of the insulator.

**Figure 2 sensors-24-01359-f002:**
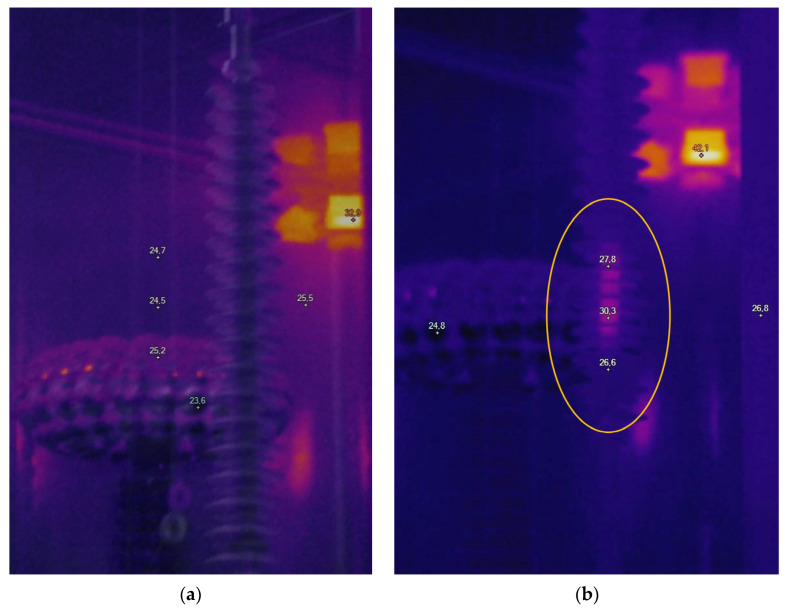
Thermographic images of two composite insulators: (**a**) a new insulator with no defects—the temperature along its axis is uniform; (**b**) a defective insulator, as evidenced by the region between the 3rd and the 7th shed and enclosed by the orange circle, which is at a higher temperature than the rest of the insulator.

**Figure 3 sensors-24-01359-f003:**
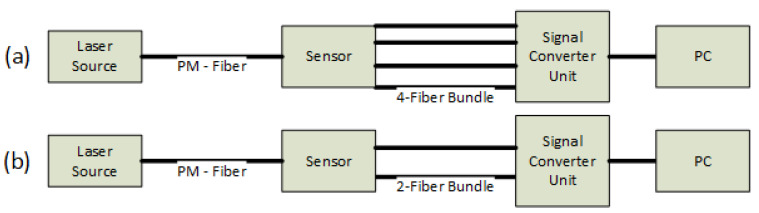
Comparison of two setups of the optic E-field sensor: (**a**) 4-fiber bundle configuration—it can measure two components of the electric field; (**b**) 2-fiber bundle configuration (this work setup)—it can measure only one component.

**Figure 4 sensors-24-01359-f004:**
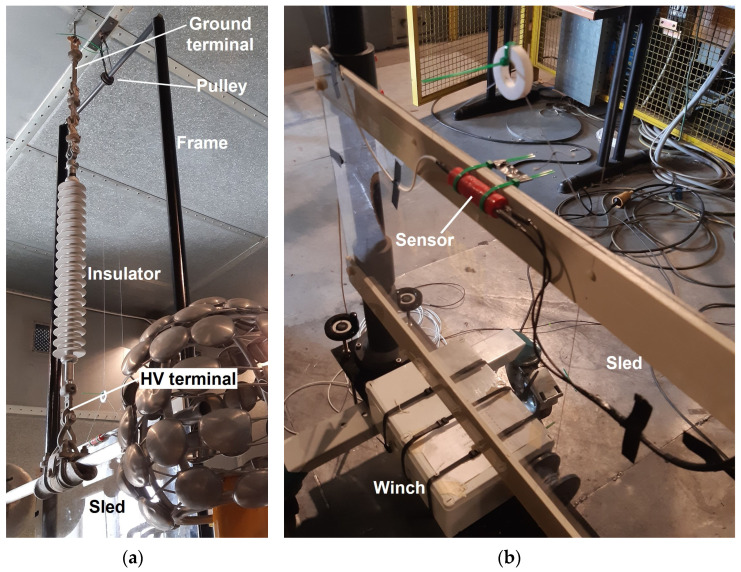
Experimental setup: (**a**) overview; (**b**) detail of the sensor and the winch.

**Figure 5 sensors-24-01359-f005:**
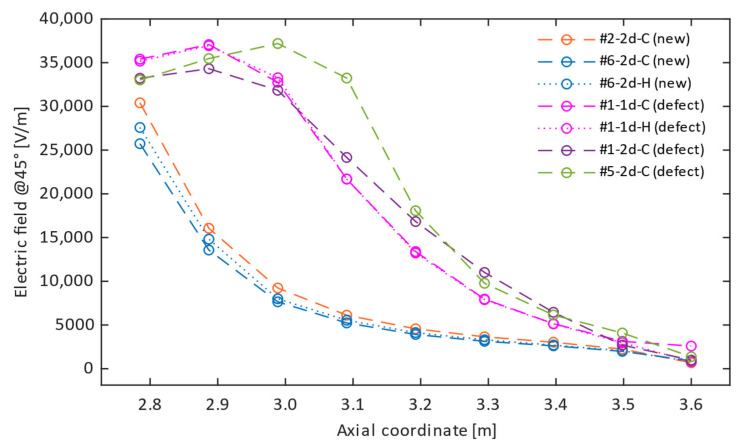
Experimental profiles of the 45° component of the electric field for the tested insulators.

**Figure 6 sensors-24-01359-f006:**
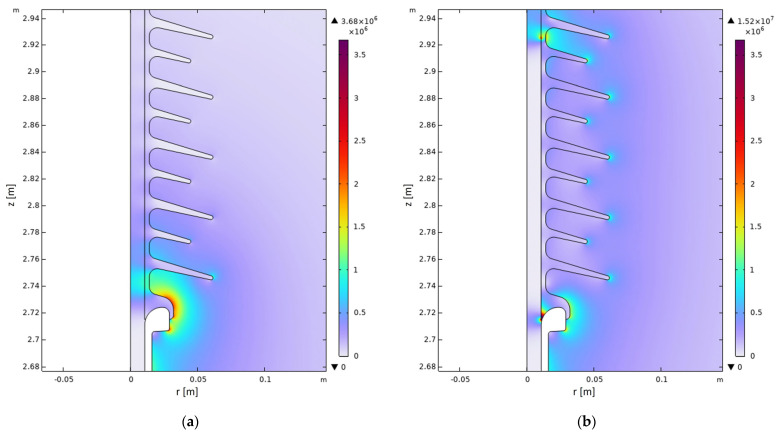
Zoom-in view of the simulated distribution of the electric field (modulus) generated by an insulator for an axisymmetric geometry: (**a**) electric field distribution of a defect-free insulator; (**b**) electric field distribution of an insulator that has a 0.1 mm-thick delamination between the fiberglass rod and the silicone housing spanning from the lower end-fitting to the 4th–5th shed for a total height of 210 mm and having the electric properties (*σ_d_* and *ε_r,d_*) of water.

**Figure 7 sensors-24-01359-f007:**
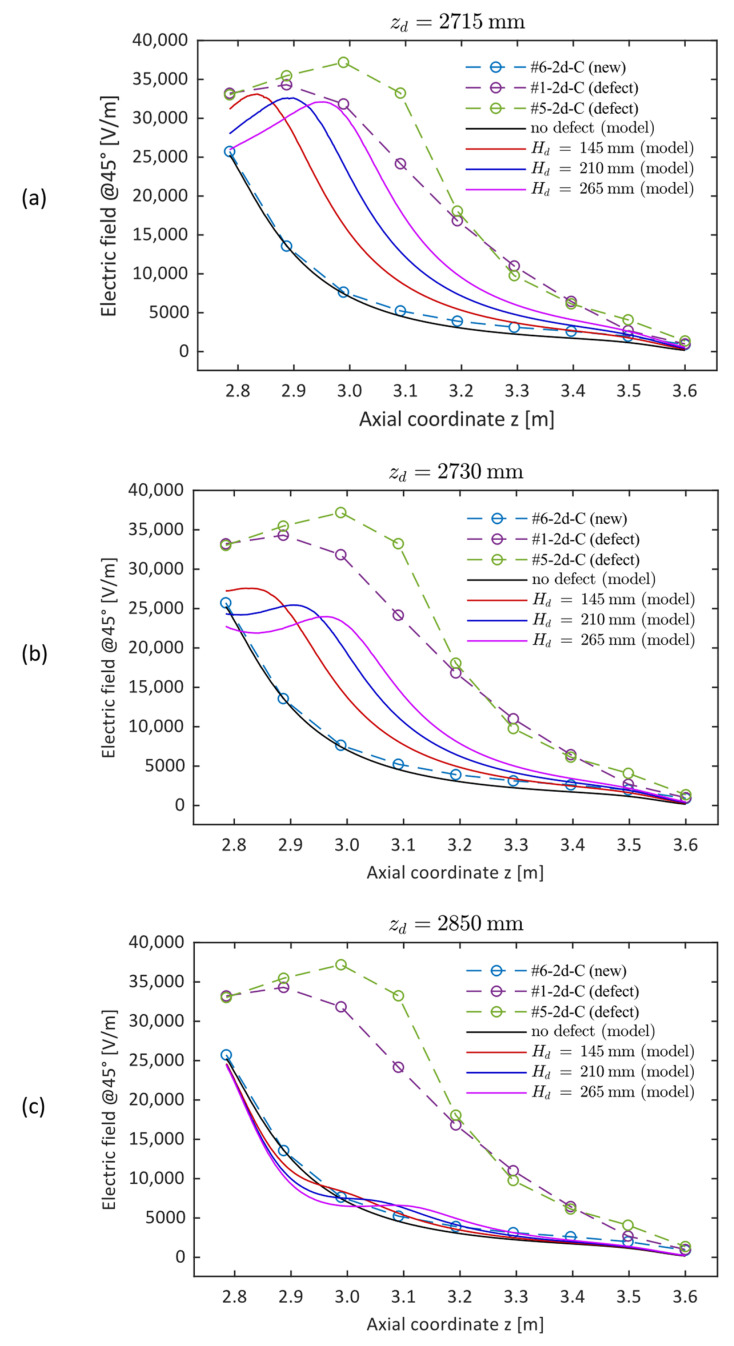
Electric field profiles obtained from the finite element model for different dimensions of the defect (solid lines) and for three different defect locations (**a**–**c**) compared with the profiles obtained experimentally. All three graphs also show the simulated profile for a new insulator (solid black line): (**a**) defect positioned at the HV terminal; (**b**) defect positioned 15 mm above the HV terminal; (**c**) defect positioned 135 mm above the HV terminal.

**Figure 8 sensors-24-01359-f008:**
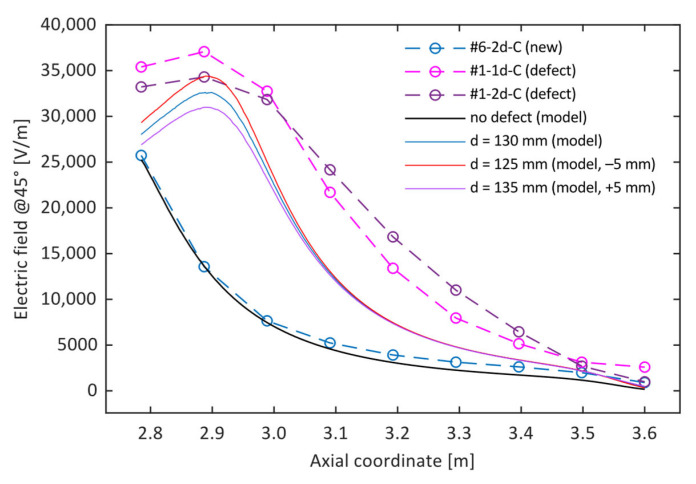
Electric field profiles obtained from the finite element model for a specific defect geometry (*z_d_* = 2715, *H_d_* = 210 mm) at three different distances from the insulator axis.

**Figure 9 sensors-24-01359-f009:**
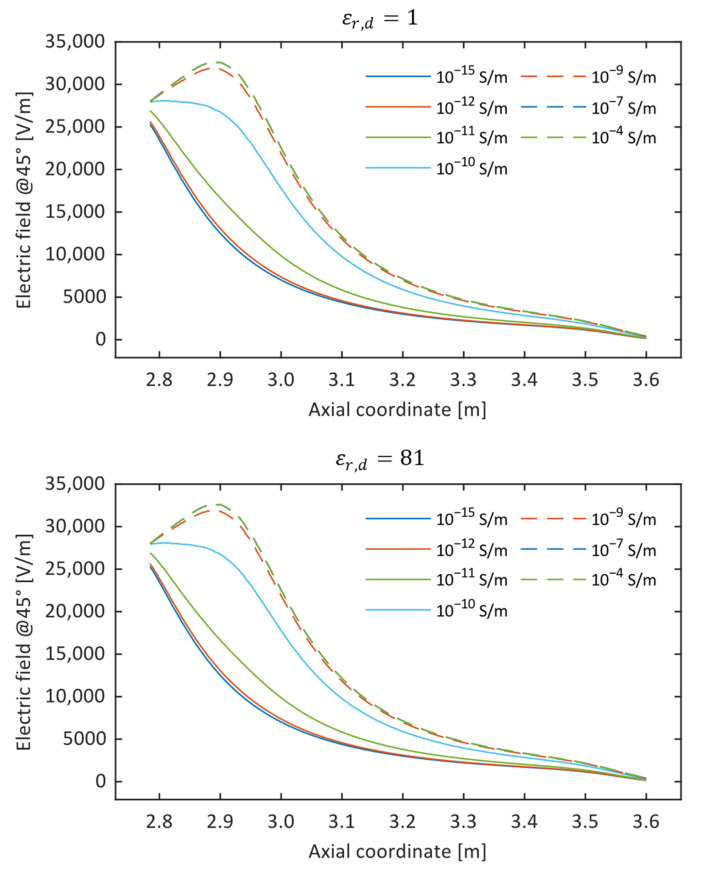
Simulated electric field profiles for a given defect geometry (*z_d_* = 2715, *H_d_* = 210 mm) for different values of the defect conductivity and for two extreme values of its permittivity.

**Figure 10 sensors-24-01359-f010:**
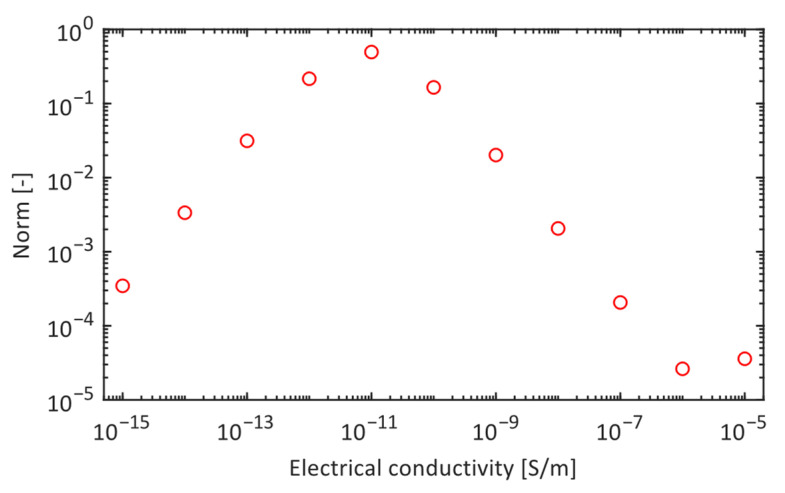
Measurement of the relative distance between electric field profiles as the conductivity of the defect increases (*z_d_* = 2715, *H_d_* = 210 mm).

**Table 1 sensors-24-01359-t001:** Test conditions and IDs.

Insulator ID	Insulator Condition	Test Day	Test Conditions	Test ID
#1	Defected	d1	Cold	#1-d1-C
		d1	Hot	#1-d1-H
		d2	Cold	#1-d2-C
#2	New	d2	Cold	#2-d2-C
#5	Defected	d2	Cold	#5-d2-C
#6	New	d2	Cold	#6-d2-C
		d2	Hot	#6-d2-H

**Table 2 sensors-24-01359-t002:** Relative permittivity and electrical conductivity of insulator materials and of air.

Parameter		Value	Description
$\varepsilon_{r,FRP}$	[-]	6.5	Relative permittivity of the rod
$\sigma_{FRP}$	[S/m]	2.00 × 10^−15^	Electrical conductivity of the rod
$\varepsilon_{r,sil}$	[-]	4.3	Relative permittivity of silicone
$\sigma_{sil}$	[S/m]	5.00 × 10^−14^	Electrical conductivity of silicone
$\varepsilon_{r,air}$	[-]	1	Relative permittivity of air
$\sigma_{air}$	[S/m]	3.00 × 10^−15^	Electrical conductivity of air

**Table 3 sensors-24-01359-t003:** Geometrical parameters of the simulated defect and their range of sweep.

Parameter		Min	Max	Description
$r_{d}$	[mm]	10.5		Radial position of the defect
$z_{d}$	[mm]	2715	2850	Axial position of the lower end of the defect
$H_{d}$	[mm]	145	265	Height of the defect

**Table 4 sensors-24-01359-t004:** Range of change in conductivity and permittivity of the defect.

Parameter		Min	Max	Description
$\varepsilon_{r,d}$	[-]	1	81	Relative permittivity of the defect
$\sigma_{d}$	[S/m]	1 × 10^−15^	1 × 10^−4^	Electrical conductivity of the defect

## Data Availability

The data presented in this study are available upon request from the corresponding author.

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
