# Peer review of "Diagnostics of Internal Defects in Composite Overhead Insulators Using an Optic E-Field Sensor"

_sensors, 2024, doi:10.3390/s24051359_

Round 1
Reviewer 1 Report
Comments and Suggestions for Authors
interesting work and valuable results.
but your work needs an extension on the influence of pollution too.
Author Response
Please, see the attached file.

Reviewer 2 Report
Comments and Suggestions for Authors
The paper titled” Diagnostics of composite overhead insulators using an optic E-field sensor” provides an application of optic E-field sensor for internal discharge detection of composite insulators. It contains enough quality to be accepted to Sensors after revising the following comments and some editorial points.
1. The reviewer suggested the authors to rethink the title. Firstly, “Diagnostics of composite overhead insulators” is a bit broad. The function of the optic E-field sensor used in this paper is to detect the electric field distortion of internal defects. A more specific title would be better.
2. The reviewer is interested that how shall the sensor to be installed onsite, on the insulators of the transmission towers.
3. A zoom-in subfigure of the defect area in Figure 6 would be better.
Comments on the Quality of English LanguageThe English is good but can be improved by making the description more logical, especially in section 3.2. The description of how the FEM model is established belongs to methodology, so it should be included in chapter 2.
Author Response
Please, see the attached file

Reviewer 3 Report
Comments and Suggestions for Authors
This article deals with an optical electric field sensor that was employed to analyze internal flaws within a series of composite insulators, with applications of finite element parameter comparison done from outcomes of the experimental setup. The following suggestions to be considered before further processing.
The motto of composite insulator testing to be added in the introduction section for goodness of the fresh readers.
Identified research challenges in the field available methods are missing the state of art.
Proposed sensor-based measurement system performance enrichment values (numerical values) to be listed in the abstract section.
List the experimental setup results variations due to 4 fiber bundle and 2 fiber bundles to some extent.
Clearly mentioned the test conditions parameters and assumptions about geometry models during the COMSOM tools are to be addressed in the right section of the article
Comments on the Quality of English LanguageModerate editing of English language required
Author Response
Please, see the attached file
